# Revocable Attribute-Based Encryption with Efficient and Secure Verification in Smart Health Systems

Zhou Chen, Lidong Han * and Baokun Hu

School of Information Science and Technology, Hangzhou Normal University, Hangzhou 311121, China;
chenzhou@stu.hznu.edu.cn (Z.C.); bourne@hznu.edu.cn (B.H.)
* Correspondence: ldhan@hznu.edu.cn

**Abstract:** By leveraging Internet of Things (IoT) technology, patients can utilize medical devices to upload their collected personal health records (PHRs) to the cloud for analytical processing or transmission to doctors, which embodies smart health systems and greatly enhances the efficiency and accessibility of healthcare management. However, the highly sensitive nature of PHRs necessitates efficient and secure transmission mechanisms. Revocable and verifiable attribute-based encryption (ABE) enables dynamic fine-grained access control and can verify the integrity of outsourced computation results via a verification tag. However, most existing schemes have two vital issues. First, in order to achieve the verifiable function, they need to execute the secret sharing operation twice during the encryption process, which significantly increases the computational overhead. Second, during the revocation operation, the verification tag is not updated simultaneously, so revoked users can infer plaintext through the unchanged tag. To address these challenges, we propose a revocable ABE scheme with efficient and secure verification, which not only reduces local computational load by optimizing the encryption algorithm and outsourcing complex operations to the cloud server, but also updates the tag when revocation operation occurs. We present a rigorous security analysis of our proposed scheme, and show that the verification tag retains its verifiability even after being dynamically updated. Experimental results demonstrate that local encryption and decryption costs are stable and low, which fully meets the real-time and security requirements of smart health systems.

**Keywords:** attribute-based encryption; revocable; verifiable; smart health; personal health records

**MSC:** 68M25

## 1. Introduction

The rapid development of the Internet of Things (IoT) and cloud computing has significantly enhanced healthcare management [1–3]. On the one hand, IoT devices, such as smartwatches, enable real-time health data capture and analysis, which assists in disease prevention. On the other hand, patients upload their personal health records (PHRs) to the cloud, which enable doctors to accurately assess patients' health status and deliver precise remote treatments. Such integration ensures precision and accessibility in modern medical care.

The sensitive nature of PHRs demands rigorous access control. Attribute-based encryption (ABE) provides a cryptographic solution that regulates data accessibility based on the attributes of recipients. For instance, after measuring blood glucose levels with a home blood glucose meter, a diabetic patient can execute ABE to encrypt the data

with an access policy $\mathcal{T} = ($"endocrinologist" $\vee$ "cardiologist"$)$, which ensures that only endocrinologists or cardiologists can access the data. Specifically, endocrinologists analyze the glucose data to evaluate the patient's diabetes control status, while cardiologists use the data to assess the patient's cardiovascular risks. However, in real-world scenarios, access control for PHRs must dynamically adapt to the progression of a patient's condition rather than remaining static. Consider a situation where the patient's condition deteriorates during subsequent blood sugar tests. To ensure professional analysis and precise treatment, the system must revoke access privileges for non-specialists and update the policy from $\mathcal{T}$ to $\mathcal{T}' = ($"endocrinologist" $\vee$ "cardiologist"$) \wedge$ "certified specialist". This demonstrates the critical need for revocable attribute-based encryption (RABE) schemes capable of efficiently implementing policy updates.

Considering the limited computational resources of IoT medical devices, it is essential to reduce local processing overhead to ensure rapid information transmission [4]. While most ABE schemes outsource complex computations to cloud servers, this approach introduces security risks where malicious servers might leak private data or tamper with results [5,6]. To mitigate these risks, strict control over outsourced data is critical. Cloud servers must be prevented from obtaining plaintext information or valid decryption keys from outsourced data, and the integrity of their computational outputs must be rigorously verified. Ge et al. [7] proposed a RABE solution, which uses a verification tag computed from the private message $m$ and an auxiliary random element $m'$. This tag enables decryptors to verify the integrity of outsourced results. Although this approach achieves verifiability, it doubles the computational workload by encrypting both $m$ and $m'$ using identical algorithms. Furthermore, during revocation, it is imperative that all components within the original ciphertext be refreshed to prevent revoked users from extracting useful information by exploiting unmodified components. However, existing schemes fail to update the tag during revocation. Since the tag remains bound to the private message $m$, revoked users could infer plaintext by correlating different ciphertext versions via the unchanged tag. Therefore, how to enable the cloud server to update the verification tag during revocation without accessing the plaintext and ensure the new tag can still effectively validate data integrity remains a critical unsolved challenge.

### 1.1. Motivations and Contributions

As discussed above, due to the need for two secret sharing operations during encryption and the failure to update the verification tag upon revocation, most of existing schemes suffer from inefficient computation and insufficient security when ensuring data integrity. In order to provide an efficient and secure access control solution for smart health scenarios, we propose a novel RABE scheme that achieves efficient data verification, and strict ciphertext update. The contributions of this work are summarized as follows:

- First, to achieve integrity verification, we introduce an auxiliary random message $m'$. Through ingenious computations, we ensure that the entire encryption process only requires executing the secret sharing operation once, which reduces computational overhead on IoT medical devices and makes the verification mechanism more efficient. Furthermore, in the access structure, we utilize the linear integer secret sharing (LISS) that operates over integers and achieves lower secret sharing overhead, which we experimentally demonstrate in Appendix A.
- Second, during the revocation process, both the verification tag and the access policy are updated, which ensures that ciphertexts are fully updated and prevents revoked users from accessing any useful information from previous ciphertexts.
- Third, the decryption key is split into a user's key and a corresponding cloud server's key, which allows partial decryption task to be outsourced to the cloud server. In addi-

tion, by outsourcing revocation and partial encryption procedures to the cloud server, we minimize the local computational load on IoT medical devices.

### 1.2. Related Work

*Attribute-Based Encryption.* As a significant cryptographic primitive, ABE was initially proposed by Sahai and Waters [8]. ABE can be broadly categorized into two primary forms: Key-Policy Attribute-Based Encryption (KP-ABE) and Ciphertext-Policy Attribute-Based Encryption (CP-ABE). In KP-ABE, access policies are embedded in users' keys, while ciphertexts are associated with sets of attributes. Decryption is only possible if the attributes in the ciphertext satisfy the access policy defined in the user's key. This makes KP-ABE particularly useful in scenarios where an authority controls user access, such as pay-TV systems [9]. Conversely, in CP-ABE, users' keys are linked to attribute sets, and ciphertexts are tied to access policies. The ability of CP-ABE to allow data owners to autonomously define access policies makes it highly advantageous for cloud data sharing [10]. So far, many research studies have focused on optimizing ABE performance, with emphasis on enhancing security [11–14] and improving efficiency [15–21]. Meanwhile, Wan et al. [22] extended ABE with a hierarchical user structure. Their scheme not only enhances scalability but also retains the flexibility and fine-grained access control capabilities of ABE. Chase et al. [23] extended ABE to a multi-authority model, where multiple authorities can issue user attributes. In their subsequent work [24], they enhanced the model's resilience against pooling attack among multiple authorities, which improved both security and privacy. Most ABE schemes employ access structures based on linear secret sharing schemes (LSSSs), where secret sharing is computed over finite fields. In 2006, Damgard and Thorbek first proposed LISSs [25], in which secret sharing is computed over integers, so the overhead of secret sharing is lowered. Devevey et al. [26] established that LISS possesses small linear reconstruction coefficients, which enables the multiplication of decryption shares without significantly amplifying underlying noise terms. Balu et al. [21] designed a CP-ABE scheme implemented based on LISSs, through which their scheme enables an effective expression of the access policy and efficient sharing of the secret exponent. Zhao et al. [27] proposed a decentralized attribute-based access control scheme with fine-grained policy enforcement, where linear integer secret sharing (LISS) enables secure symmetric key distribution across multiple attributes while maintaining cryptographic security guarantees.

*Revocable.* When users exit the system or their attributes change, their access permissions to the data must be updated. Therefore, the revocation mechanism is particularly important [28]. Traditional revocation methods include indirect revocation and direct revocation [29]. Indirect revocation typically involves periodically distributing updated keys to all non-revoked users. If users do not receive the updated keys, this indicates that they have been revoked, as seen in schemes [30–32]. However, this method has two significant drawbacks. First, updating keys is time-consuming. Second, since keys are updated periodically, real-time revocation cannot be achieved. Direct revocation methods embed a real-time updated revocation list into the ciphertext, which enables real-time revocation, as seen in schemes [33–35]. However, a significant drawback of this approach is that the revocation list continues to expand over time. This growth leads to significant storage overhead and may eventually become an efficiency bottleneck for the scheme. Liu et al. [36] introduced a revocable CP-ABE scheme, which integrates both direct and indirect revocation approaches. This scheme achieves both real-time revocation and effective control over the length of the revocation list. However, this approach is unable to handle ciphertext update. When the data's access policy changes, a revocation operation must be performed. In CP-ABE, revocation caused by such a reason can be implemented by updating the access policy embedded in the ciphertext. Jiang et al. [37] designed a

revocable CP-ABE scheme, which can support the AND gate on attributes. On the other hand, Ge et al. [7] proposed a new revocable CP-ABE approach. By representing access policy updates as transformations in a matrix structure, the scheme makes it easier for readers to understand the principles of access policy updates and enhances the scheme's scalability. Subsequently, Chen et al. [38] utilized the same method to implement revocation and further achieved constant decryption overhead while attaining full security.

*Verifiable.* Owing to the limited local computational resources of IoT medical devices, complex operations need to be outsourced to cloud servers. However, since cloud servers may be malicious, a verification mechanism is indispensable. In the approaches designed by Ge et al. [7] and Chen et al. [38], a verification tag is introduced. This tag is generated from the private message $m$ and an auxiliary random message $m'$, both of which are encrypted with the same method. After decryption, the decryptor reconstructs the verification tag and compares it with the tag embedded in the ciphertext to verify integrity. However, this method has two flaws. First, the addition of $m'$ doubles the computational overhead. Second, when the ciphertext is updated during revocation, the verification tag is not updated. If revoked users compare the verification tags from the previous ciphertexts with the tag in the updated ciphertext, they can infer the plaintext in the updated ciphertext. Huang et al. [39] introduced a solution to reduce computational overhead. Specifically, a symmetric key encrypts the private message $m$, and the key itself is embedded as plaintext in ABE encryption. The verification tag is derived from the symmetric key and $m$. Although this scheme imposes almost no additional computational overhead in implementing the verification mechanism, it still has drawbacks. During revocation, only the ABE ciphertext is updated, whereas the symmetric key and the ciphertext of $m$ encrypted with this key remain unchanged. As a result, revoked users can still know that their previously accessed message remains valid. Additionally, this scheme also suffers from the issue of the verification tag not being updated. Miao et al. [40] developed a verifiable ABE framework that offloads partial encryption and decryption tasks to the cloud server. To guarantee correctness, distinct validation methods are specifically designed for outsourced encryption and decryption processes. Nevertheless, their approach lacks revocation support.

## 2. Preliminaries

### 2.1. Bilinear Mapping

Let $\mathbb{G}$ and $\mathbb{G}_T$ denote cyclic groups of prime order $p$, with $g$ as a generator of $\mathbb{G}$. A function $\hat{e} : \mathbb{G} \times \mathbb{G} \to \mathbb{G}_T$ is defined as a bilinear pairing if it satisfies the following properties:

- For all $g_1, g_2 \in \mathbb{G}$ and $a, b \in \mathbb{Z}_p^*$, the equality $\hat{e}(g_1^a, g_2^b) = \hat{e}(g_1, g_2)^{ab}$ holds.
- There exists $g_1, g_2 \in \mathbb{G}$ such that $\hat{e}(g_1, g_2) \neq 1_{\mathbb{G}_T}$, where $1_{\mathbb{G}_T}$ denotes the identity element in $\mathbb{G}_T$.
- The value $\hat{e}(g_1, g_2)$ can be efficiently computed for any $g_1, g_2 \in \mathbb{G}$ in polynomial time.

### 2.2. Target Collision-Resistant Hash Function

A hash function $H$ is termed target collision-resistant (TCR) if no probabilistic polynomial-time (PPT) adversary can generate a collision for a randomly chosen input. Specifically, given a uniformly sampled element $x \in D$, where $D$ denotes the domain of $H$, any efficient adversary $\mathcal{A}$ succeeds in finding a distinct $y \in D$ (i.e., $y \neq x$) such that $H(y) = H(x)$ only with negligible probability.

Formally, for all PPT adversaries $\mathcal{A}$, the following advantage is negligible in $\lambda$:

$$Adv_{\mathcal{A}}^{\mathsf{TCR}} = \Pr\left[y \neq x \wedge H(y) = H(x) \mid x, y \in D\right].$$

### 2.3. Complex Assumptions

**Definition 1** (q-parallel BDHE assumption)**.** *Let $\mathbb{G}$ and $\mathbb{G}_T$ denote cyclic groups of prime order p, with g as a generator of $\mathbb{G}$. Random elements $a, r, b_1, \ldots, b_q \in \mathbb{Z}_p^*$ are selected. Given the following challenge instance:*

$$
\Delta = \begin{cases}
g, g^r, g^a, \ldots, g^{a^q}, g^{a^{q+2}}, \ldots, g^{a^{2q}}, \\
\forall_{1 \le j \le q} \; g^{r \cdot b_j}, g^{a/b_j}, \ldots, g^{a^q/b_j}, g^{a^{q+2}/b_j}, \\
\forall_{1 \le j,k \le q, k \ne j} \; g^{a \cdot r \cdot b_k/b_j}, \ldots, g^{a^q \cdot r \cdot b_k/b_j}
\end{cases}
\tag{1}
$$

*the adversary $\mathcal{A}$ must distinguish $\hat{e}(g,g)^{a^{q+1}r}$ from a random element $T \in \mathbb{G}_T$. The q-parallel BDHE assumption asserts that for any PPT adversary $\mathcal{A}$, the advantage*

$$
Adv_{\mathcal{A}} = \left| \Pr\left[ \mathcal{A}(\Delta, e(g,g)^{a^{q+1}r}) = 1 \right] - \Pr[\mathcal{A}(\Delta, T) = 1] \right|
$$

*is negligible in $\lambda$.*

**Definition 2** (Discrete Logarithm (DL) Assumption)**.** *Given a tuple $(\hat{e}, \mathbb{G}, \mathbb{G}_T, p, g, g^{\delta})$, where $(\hat{e}, \mathbb{G}, \mathbb{G}_T, p, g)$ defines a bilinear pairing, $g \in \mathbb{G}$, and $\delta \in \mathbb{Z}_p^*$, the DL assumption indicates that no PPT adversary $\mathcal{A}$ can compute $\delta$ with non-negligible advantage. Formally, the advantage of $\mathcal{A}$, $\Pr[\mathcal{A}(\hat{e}, \mathbb{G}, \mathbb{G}_T, p, g, g^{\delta}) = \delta]$, is negligible in $\lambda$.*

### 2.4. Linear Integer Secret Sharing (LISS)

In the LISS scheme [25], a secret integer is chosen from a public interval. Shares are generated as linear combinations of the secret value and random integers selected by the dealer. The secret can be reconstructed by computing an integer linear combination of shares from any authorized set.

Let $P = \{1, \ldots, n\}$ represent $n$ shareholders, $D$ the dealer, $\Gamma$ the access structure on $P$, and $s$ the secret selected from the public interval $[0, 2^l]$. The dealer $D$ distributes $s$ to participants in $P$ according to $\Gamma$. Any authorized set $A \in \Gamma$ can recover $s$, while unauthorized sets $A \notin \Gamma$ cannot. To implement this, $D$ constructs the following LISS scheme:

First, $D$ generates a $d \times e$ distribution matrix $M \in \mathbb{Z}^{d \times e}$ and a distribution vector $\vec{\rho} = (s, \rho_2, \ldots, \rho_e)^T$, where each $\rho_i$ is uniformly and randomly chosen from $[0, 2^{l_0+k}]$. Here, $l_0$ is a predefined constant and $k$ is the security parameter. $D$ computes the shares as

$$
M \cdot \vec{\rho} = (s_1, \ldots, s_d)^T,
$$

where $s_i$ is the $i$-th share. A surjective function $\psi : \{1, \ldots, d\} \to P$ assigns $s_i$ to shareholder $\psi(i)$, who holds the $i$-th row of $M$.

**Definition 3.** *The LISS scheme is correct if, for any authorized set $A \in \Gamma$, there exists a set of coefficients $\{\lambda_i\}$ such that $s = \sum_{i \in A} \lambda_i s_i$, then the LISS is correct.*

**Definition 4.** *The LISS is private if, for any unauthorized set $A \notin \Gamma$ and any two secrets $s, s'$, the distribution of shares generated from s is statistically indistinguishable from the distribution generated from $s'$.*

### 2.5. Integer Span Program (ISP)

Let $M \in \mathbb{Z}^{d \times e}$ be a $d \times e$ integer matrix, $\psi : \{1, \ldots, d\} \to P$ be a surjective function, and $\vec{\epsilon} = (1, 0, \ldots, 0)^T \in \mathbb{Z}^d$ be the target vector. The function $\psi$ maps rows of $M$ to participants $p$, and each participant may hold multiple rows.

**Definition 5** (Integer Span Program). *For any set $A \subset P$, the quadruple $\mathfrak{M} = (\mathbb{Z}, M, \psi, \epsilon)$ is an Integer Span Program (ISP) under the access structure if the following condition holds:*

- *If $A \in \Gamma$, there exists a vector $\vec{\lambda}_A = (\lambda_1, \ldots, \lambda_d)^T \in \mathbb{Z}^d$ such that $M_A^T \cdot \vec{\lambda}_A = \vec{\epsilon}$, where $\vec{\lambda}_A$ is called the spanning vector of $A$.*
- *If $A \notin \Gamma$, there exists a vector $\vec{k}_A = (k_1, \ldots, k_e)^T \in \mathbb{Z}^e$ such that $M_A \cdot \vec{k}_A = 0$, where $k_i = 1$, and $k_i$ is defined as the elimination vector of $A$.*

If we have an ISP $\mathfrak{M} = (\mathbb{Z}, M, \psi, \epsilon)$, which computes $\Gamma$, we build an LISS scheme for $\Gamma$ as follows: we use $M$ as the distribution matrix and $l_0 = l + \lceil \log_2(k_{\max}(e-1)) \rceil + 1$, where $l$ is the length of the secret and $k_{\max} = \max\{|a| \mid a \text{ is an entry in some sweeping vector}\}$.

According to Definition 5, for an authorized set $A$, the secret $s$ can be reconstructed using

$$s_A^T \cdot \vec{\lambda}_A = (M_A \cdot \vec{p})^T \cdot \vec{\lambda}_A = \vec{\rho}^T \cdot (M_A^T \cdot \vec{\lambda}_A) = \vec{\rho}^T \cdot \vec{\epsilon} = s.$$

## 3. System Architecture and Definitions

### 3.1. System Architecture

As illustrated in Figure 1, our proposed scheme comprises four entities:

- Key Generation Center (KGC): In smart health systems, the KGC generates and distributes keys for all participants, including IoT medical devices, cloud servers, and doctors.
- Cloud Server (CS): The CS is tasked with storing ciphertexts and assisting in complex revocation operations and partial encryption/decryption operations.
- Data Owner (DO): DOs are IoT medical devices that collect PHRs, perform local encryption operations, and upload ciphertexts to the CS. Additionally, when the access policy of the data changes, the DO sends a revocation delegation to the CS.
- Data User (DU): Doctors act as DUs. They submit decryption requests to the CS based on their attributes, decrypt intermediate ciphertexts from the CS using their private keys and perform the treatment procedures according to the data content.

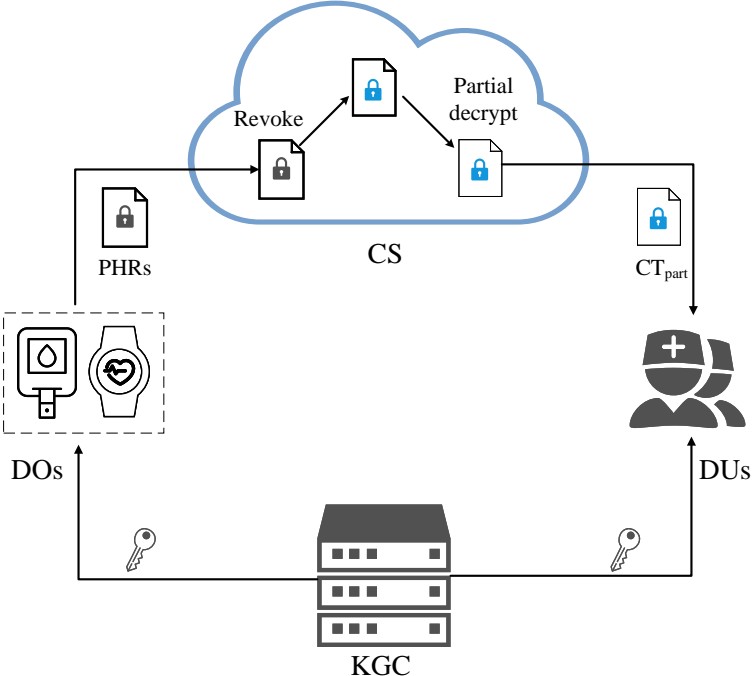

**Figure 1.** System architecture.

### 3.2. Threat Model

We consider the following adversarial threats in our system:

- Confidentiality Adversary: This category represents any malicious entity that aims to compromise data confidentiality. Without valid decryption keys, this adversary attempts to decrypt ciphertexts and access sensitive messages.
- Integrity Adversary: This adversary is typically a semi-honest cloud server that aims to violate data integrity by generating incorrect computational results during outsourced operations.

### 3.3. Scheme Definition

**Definition 6.** *Our scheme comprises the following algorithms and Figure 2 illustrates the complete interaction process between the four participating entities, including the KGC, DO, DU and CS, during the execution of these algorithms. Note that, for simplicity, we have removed the system parameters pp from the algorithm's input.*

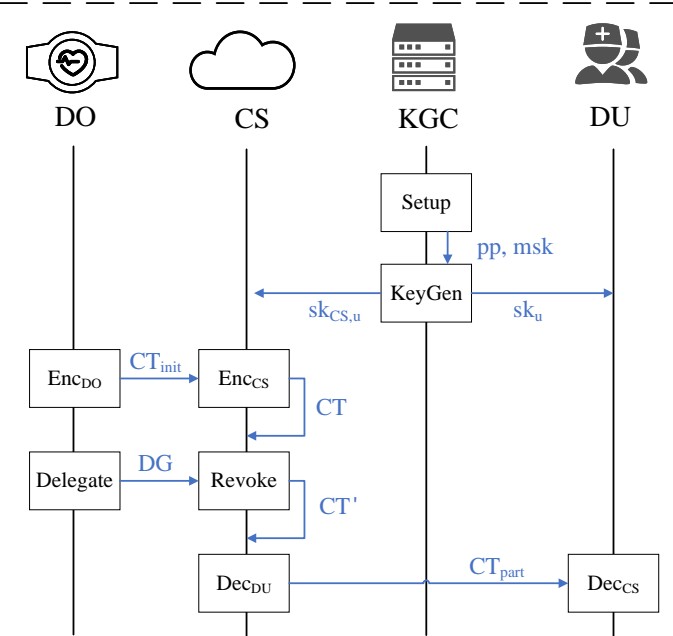

**Figure 2.** Framework of our scheme with the entities' interaction flow.

(1) $Setup(\lambda, U)$: *Inputting the security parameter $\lambda$ and attribute universe $U$, the KGC generates public parameters pp and master secret key msk.*

(2) $KeyGen(msk, S_u)$: *Given the master secret key msk and a user's attribute set $S_u$, the KGC produces the user's private key $sk_u$ and the corresponding CS's private key $sk_{CS,u}$. These keys are distributed to the user and cloud server, respectively.*

(3) $Enc(m, (M, \tau))$: *This algorithm contains two steps. First, the DO performs local encryption ($Enc_{DO}$) by processing message $m \in \mathbb{G}_T$ with access policy $(M, \tau)$ to create initial ciphertext $CT_{init}$. The CS then executes outsourced encryption ($Enc_{CS}$) to generate final ciphertext CT.*

(4) $Delegate((\widetilde{M}, \widetilde{\tau}), m, z, r)$: *This takes as input the access policy $(\widetilde{M}, \widetilde{\tau})$ to be revoked. The DO generates a revocation delegation DG using the message $m$ and parameters $z, r$ from $Enc_{DO}$, and sends DG to the CS.*

(5) $Revoke(CT, DG)$: *Given the initial CT and revocation delegation DG, the CS constructs the revoked access policy $(M', \tau')$ and outputs the revoked ciphertext $CT'$ based on this policy.*

(6) $Dec(CT', S_{DU}, sk_{CS,DU}, sk_{DU}, tag_{DG})$: *The decryption process comprises two sequential phases. First, the cloud server performs partial decryption ($Dec_{CS}$) by verifying if the attribute set $S_{DU}$ satisfies $(M', \tau')$ using inputs $CT'$, $sk_{CS,DU}$, and $S_{DU}$. If the verification*

*succeeds, it generates the partial decryption result $CT_{part}$ and sends it to the DU. The DU then executes final decryption ($Dec_{DU}$) with $CT_{part}$, $sk_{DU}$, and $tag_{DG}$, verifying the equivalence between the tag in $CT_{part}$ and DG. If the tags match, the user computes m and validates $tag' = \Phi^{H(m)}\Psi^{H(m')}$. If valid, output m; otherwise, $\perp$ is returned.*

### 3.4. Security Definitions

The security of our scheme requires selective security and data integrity.

**Definition 7** (Semantic Security). *The proposed scheme satisfies semantic security in the selective model under the condition that, for any PPT adversary $\mathcal{A}$, its advantage $Adv_{\mathcal{A}}^{Semantic}(\lambda)$ is negligible. The security game proceeds as follows:*

*(1)    Init. The adversary $\mathcal{A}$ submits a challenge access policy $(M_{d^* \times e}^*, \tau^*)$, where $d^*, e^* \leq q$.*

*(2)    Setup. The challenger $\mathcal{C}$ executes the Setup algorithm to produce pp as public parameters and msk as the master secret key, then sends pp to $\mathcal{A}$.*

*(3)    Phase I. $\mathcal{A}$ adaptively issues private key queries for attribute sets $S_u$ that do not satisfy $(M^*, \tau^*)$. $\mathcal{C}$ generates and returns the corresponding private keys.*

*(4)    Challenge. $\mathcal{A}$ provides two messages $m_0, m_1$ of equal length. $\mathcal{C}$ randomly selects $b \in \{0,1\}$, encrypts $m_b$ under $(M^*, \tau^*)$ to generate the challenge ciphertext $CT^*$, and delivers $CT^*$ to $\mathcal{A}$.*

*(5)    Phase II. $\mathcal{A}$ proceeds to issue key queries under the same restrictions as in Phase I.*

*(6)    Guess. $\mathcal{A}$ outputs $b' \in \{0,1\}$. The advantage of the adversary $\mathcal{A}$ is quantified as*

$$Adv_{\mathcal{A}}^{IND\text{-}CPA}(\lambda) = \left| \Pr[b' = b] - \frac{1}{2} \right|.$$

**Definition 8** (Integrity). *The proposed scheme satisfies data integrity provided that, for any PPT adversary $\mathcal{A}$, the advantage $Adv_{\mathcal{A}}^{DI}(\lambda)$ is negligible. The security game proceeds as follows:*

*(1)    Setup: The challenger $\mathcal{C}$ performs the Setup algorithm to create the public parameters pp and the master secret key msk, then sends pp to $\mathcal{A}$.*

*(2)    Phase I: $\mathcal{A}$ adaptively issues private key queries for any attribute set $S_u$. $\mathcal{C}$ generates and returns the corresponding private keys.*

*(3)    Challenge: $\mathcal{A}$ selects an access policy $(M, \tau)$ and a message m. $\mathcal{C}$ generates an initial ciphertext $CT_{init}$ and returns it. In revocation scenarios, $\mathcal{C}$ additionally generates and sends a revocation delegation DG.*

*(4)    Phase II: $\mathcal{A}$ proceeds to issue queries as in Phase I.*

*(5)    Output: $\mathcal{A}$ outputs a partial decryption result $CT_{part}$ and corresponding verification tags tag or tag'. The adversary wins if either of the following apply:*

**Case 1** (no revocation). *tag = tag but the decrypted message $m \neq m$;*

**Case 2** (with revocation). *$tag' = tag$ but the decrypted message $m \neq m$.*

*The advantage of the adversary $\mathcal{A}$ is defined as*

$$Adv_{\mathcal{A}}^{DI}(\lambda) = \Pr[\mathcal{A} \text{ wins}].$$

## 4. Construction

*(1)    Setup$(\lambda, S)$: The algorithm generates a bilinear pairing tuple $\mathbb{BG} = (\hat{e}, \mathbb{G}, \mathbb{G}_T, g, p)$ and selects random elements $f_1, f_2, \cdots, f_{|U|}, \Phi, \Psi \in \mathbb{G}, a, \alpha, \alpha_1, \alpha_2 \in \mathbb{Z}_p^*$, where $\alpha = \alpha_1 + \alpha_2$*

along with a hash function $H : \mathbb{G}_T \to \mathbb{Z}_p^*$. Then, KGC computes $v = \hat{e}(g,g)^{\alpha}$ and outputs the public parameters $pp$ as well as the master secret key $msk$ as follows:

$$pp = (\mathbb{BG}, \Phi, \Psi, v, g^a, H, \{f_x\}_{x \in U}),$$
$$msk = (\alpha, \alpha_1, \alpha_2).$$

(2)  *KeyGen*$(msk, S_u)$: When a user $u$ registers with the system, KGC picks a random element $s \in \mathbb{Z}_p^*$ and calculates

$$K = g^{\alpha_1} g^{as}, K_0 = g^s, K_x = f_x^s,$$
$$sk_u = g^{\alpha_2} g^{as}.$$

Then, KGC transmits $sk_{CS,u} = (K, K_0, \{K_x\}_{x \in S_u})$ to the CS and sends $sk_u$ to the user $u$.

(3)  *Enc*$(m, (M, \tau))$: The encryption algorithm is structured in two steps, as detailed in Algorithm 1. Initially, the DO performs the core initial encryption $Enc_{DO}$ to convert the message $m$ into an initial ciphertext $CT_{init}$. Subsequently, the CS executes computationally intensive operations to expand $CT_{init}$ into the final ciphertext $CT$. Notably, the entire encryption process requires only a single secret sharing operation, which reduces the computational overhead. The sub-algorithms are detailed below:

    (i)  *Enc*$_{DO}(m, (M, \tau))$: Given a private message $m \in \mathbb{G}_T$ and an access policy $(M, \tau)$, where $M$ is a $d \times e$ matrix, the DO randomly selects $r \in [0, 2^l]$ and $y_j \in [0, 2^{l_0 + k}]$ for $j \in [2, e]$ to construct a vector $\vec{u} = (r, y_2, \ldots, y_e)$. Here, $k$ is the security parameter and $l_0$ is a predefined constant. The DO then computes $\lambda_i = \vec{u} \cdot M_i$ for all $i \in [1, d]$, where $M_i$ denotes the $i$-th row of $M$. Subsequently, DO randomly selects $m' \in \mathbb{G}_T, z \in \mathbb{Z}_p^*$ and calculates

$$C_0 = m \cdot v^r, \quad C_1 = g^r,$$
$$D_0 = m' \cdot v^{z+r}, \quad D_1 = g^{z+r}, \tag{2}$$
$$D_2 = g^{az}, \quad tag = \Phi^{H(m)} \Psi^{H(m')}.$$

Finally, the DO transmits the initial ciphertexts $CT_{init} = ((M, \tau), C_0, C_1, D_0, D_1, D_2, tag, \{\lambda_i\}_{i=1}^d)$ to the CS.

    (ii)  *Enc*$_{CS}(CT_{init})$: Upon receiving $CT_{init}$, CS randomly chooses $r_i \in \mathbb{Z}_p^*$ for all $i \in [1, d]$ and computes:

$$C_{2,i} = g^{a\lambda_i} f_{\tau(i)}^{-r_i}, \quad C_{3,i} = g^{r_i}, i \in [1, d]. \tag{3}$$

Then, CS stores the final ciphertext $CT = ((M, \tau), C_0, C_1, \{C_{2,i}, C_{3,i}\}_{i=1}^d, D_0, D_1, D_2, tag)$

(4)  *Delegate*$((\widetilde{M}, \widetilde{\tau}), m, z, r)$: When the access policy is updated, to prevent the verification tag from leaking any information about the message $m$, the DO updates the verification tag and issues a revocation delegation to the CS. Specifically, DO randomly selects $m'' \in \mathbb{G}_T$ and computes

$$\overline{D}_0 = m'' \cdot v^{z+r}, \quad tag' = \Phi^{H(m)} \Psi^{H(m'')},$$

then sends $DG = ((\widetilde{M}, \widetilde{\tau}), \overline{D}_0, tag')$ to the CS.

---

**Algorithm 1:** Encryption Algorithm (*Enc*)

---

**Input** :Message $m \in \mathbb{G}_T$, access policy $(M, \tau)$

**Output**:Ciphertext $CT$

1 **DO side:**

2 Randomly select $r \in [0, 2^l]$ and $y_j \in [0, 2^{l_0+k}]$ for $j \in [2, e]$ ;

3 Construct a vector $\vec{u} = (r, y_2, \ldots, y_e)$ ;

4 Randomly select $m' \in \mathbb{G}_T, z \in \mathbb{Z}_p^*$ and generate $(C_0, C_1, D_0, D_1, D_2, tag)$ by Equation (2);

5 **for** $i = 1; i \le d; i+ = 1$ **do**

6     Compute $\lambda_i = \vec{u} \cdot M_i$;

7 Set the initial ciphertext as $CT_{init} = ((M, \tau), C_0, C_1, D_0, D_1, D_2, tag, \{\lambda_i\}_{i=1}^d)$;

8 **CS side:**

9 **for** $i = 1; i \le d; i+ = 1$ **do**

10     Randomly choose $r_i \in \mathbb{Z}_p^*$;

11     Compute $(C_{2,i}, C_{3,i})$ by Equation (3);

12 Set the final ciphertext as $CT = ((M, \tau), C_0, C_1, \{C_{2,i}, C_{3,i}\}_{i=1}^d, D_0, D_1, D_2, tag)$;

13 **return** $CT$.

---

(5)   *Revoke*$(CT, DG)$: The revocation process is described in Algorithm 2. On receiving the delegation $DG$, CS first defines the revoked access policy $(M', \tau')$ as follows:

$$M' = \left( \begin{array}{c|c|c} M & -col_1 & 0 \\ \hline 0 & \multicolumn{2}{c}{\widetilde{M}} \end{array} \right), \quad \tau'(i) = \begin{cases} \tau(i) & i \le d \\ \widetilde{\tau}(i - d) & i > d \end{cases}. \tag{4}$$

Note that $col_1$ refers to the first column of $M$, and $\widetilde{M}$ is a $d' \times e'$ matrix, where $d' = d + \tilde{d}, e' = e + \tilde{e}$. For each $i \in [d + 1, d']$, the CS sets

$$C_{3,i} = 1_{\mathbb{G}}, \quad C_{4,i} = 1_{\mathbb{G}}.$$

Next, the CS selects a vector $\vec{u}' = (r', y_2', \ldots, y_{e'}')$, where $r' \in [0, 2^l]$ and $y_j' \in [0, 2^{l_0+k}]$ for $j \in [2, e']$, computes $\lambda_i' = \vec{u}' \cdot M_i'$ for all $i \in [1, d']$, and randomly chooses $z', r_i' \in \mathbb{Z}_p^*$ $(i \in [1, d'])$. The CS then calculates

$$C_0' = C_0 \cdot v^{r'}, C_1' = C_1 \cdot g^{r'},$$

$$C_{2,i}' = C_{2,i} \cdot g^{a\lambda_i'} f_{\tau'(i)}^{-r_i'}, C_{3,i}' = C_{3,i} \cdot g^{r'i}, i \in [1, d'], \tag{5}$$

$$D_0' = \overline{D}_0 \cdot v^{z'+r'}, D_1' = D_1 \cdot g^{z'+r'}, D_2' = D_2 \cdot g^{az'}.$$

Finally, the CS outputs the revoked ciphertext: $CT' = ((M', \tau'), C_0', C_1', \{C_{2,i}', C_{3,i}'\}_{i=1}^{d'}, D_0', D_1', D_2', tag')$.

(6)   *Dec*$(CT', S_{DU}, sk_{CS,DU}, sk_{DU}, tag_{DG})$: The decryption algorithm involves two phases, as outlined in Algorithm 3. Firstly, the DU submits a decryption request to the CS. Upon confirming that the DU is authorized for data decryption, the CS executes the partial decryption algorithm $Dec_{CS}$ with $sk_{CS,DU}$ to generate $CT_{part}$ and forwards it to the DU. Secondly, the DU decrypts $CT_{part}$ with $sk_{DU}$ and checks the message validity via the tag. The sub-algorithms are defined as follows:

(i) $Dec_{CS}(CT', sk_{CS,DU}, S_{DU})$: Given a ciphertext $CT'$ and a user's attribute set $S_{DU}$, the CS first verifies if $S_{DU}$ satisfies $(M', \tau')$. If not, the process terminates and outputs $\perp$. Otherwise, the CS defines $A' = \{i \mid \tau'(i) \in S_{DU}\} \subset \{1, \ldots, d'\}$ and finds constant elements $\eta'_i \in \mathbb{Z}_p^*$ such that $\sum_{i \in A'} \eta'_i \cdot M'_i = (1, 0, \ldots, 0)$. Then, the CS computes

$$W^2 = \prod_{i \in A'} (\hat{e}(K_0, C'_{2,i}) \cdot \hat{e}(K_{\tau'(i)}, C'_{3,i}))^{2\eta'_i},$$

$$C_{DU} = C'_0 / \frac{\hat{e}(C'_1, K)}{W^2}, \quad D_{DU} = D'_0 / \frac{\hat{e}(D'_1, K)}{\hat{e}(D'_2, K_0)^2 \cdot W^2}. \tag{6}$$

The CS sends the partially decrypted ciphertext $CT_{part} = (C'_1, C_{DU}, D'_1, D_{DU}, tag')$ to the DU.

(ii) $Dec_{DU}(CT_{part}, sk_{DU}, tag_{DG})$: Taking as input an intermediate ciphertext $CT_{part}$ and the verification tag $tag_{DG}$ from $DG$, the DU first verifies whether the verification tag in $CT_{part}$ is equal to $tag_{init}$. If not, the algorithm outputs $\perp$ and aborts. Otherwise, it computes

$$m = \frac{C_{DU}}{\hat{e}(C'_1, sk_{DU})}, \quad m'' = \frac{D_{DU}}{\hat{e}(D'_1, sk_{DU})}. \tag{7}$$

The DU verifies whether $\overline{tag'} = \Phi^{H(m)} \Psi^{H(m'')}$. If valid, it outputs $m$; otherwise, $\perp$ is returned.

---

**Algorithm 2:** Revoke Algorithm (*Revoke*)

**Input** : Original ciphertext $CT$, revocation delegation $DG$

**Output**: Revoked ciphertext $CT'$

1 Construct the revoked access policy $(M', \tau')$ by Equation (4);
2 Randomly select $r' \in [0, 2^l]$ and $y'_j \in [0, 2^{l_0+k}]$ for $j \in [2, e]$ ;
3 Construct a vector $\vec{u}' = (r', y'_2, \ldots, y'_{e'})$ ;
4 **for** $i = d+1; i \le d'; i+ = 1$ **do**
5     Set $C_{3,i} = 1_{\mathbb{G}}, C_{4,i} = 1_{\mathbb{G}}$ ;
6 **for** $i = 1; i \le d'; i+ = 1$ **do**
7     Compute $\lambda'_i = \vec{u}' \cdot M'_i$;
8 Randomly select $z', r'_i \in \mathbb{Z}_p^*$ for $i \in [1, d']$;
9 Compute $(C'_0, C'_1, C'_{2,i}, C'_{3,i}, D'_0, D'_1, D'_2)$ by Equation (5);
10 Set the revoked ciphertext as
    $CT' = ((M', \tau'), C'_0, C'_1, \{C'_{2,i}, C'_{3,i}\}_{i=1}^{d'}, D'_0, D'_1, D'_2, tag');$
11 **return** $CT'$.

---

**Remark 1.** *In this scheme, the partial decryption operations by the CS do not lead to privacy data leakage. First, the ciphertext component $C'_0 = m \cdot \hat{e}(g, g)^{\alpha r'}$ in $CT'$ encrypts the message $m$ using the master private key $\alpha$, where the master private key is split via $\alpha = \alpha_1 + \alpha_2$. The CS's key only holds $\alpha_1$, while $\alpha_2$ is embedded in the user's private key. Since the CS cannot reconstruct the complete $\alpha$ solely from $\alpha_1$, independent decryption is infeasible. Second, the design of the verification tag $tag' = \Phi^{H(m)} \Psi^{H(m'')}$ in $CT'$ ensures that any attempt by the CS to derive the plaintext $m$ from it would require solving the discrete logarithm problem, which is computationally infeasible under standard cryptographic assumptions. More importantly, even if the CS colludes with revoked*

*users, it remains unable to decrypt the ciphertext. During the key generation phase, the KGC independently selects random values $s_x$ and $s_y$ for revoked users (e.g., $DU_x$) and non-revoked users (e.g., $DU_y$) and generates key pairs $(sk_{CS,x}, sk_x)$ and $(sk_{CS,y}, sk_y)$, respectively. If the CS colludes with $DU_x$ and attempts to forward the intermediate ciphertext decrypted using $sk_{CS,y}$ to $DU_x$, the local decryption process will yield $\frac{m \cdot \hat{e}(g^{r+r'}, g^{\alpha_2} g^{as_y})}{\hat{e}(g^{r+r'}, g^{\alpha_2} g^{as_x})}$. Due to the mismatch between $s_x$ and $s_y$, the encryption components cannot be canceled out. This guarantees that an unauthorized decryption of the ciphertext is impossible.*

---

**Algorithm 3:** Decryption Algorithm (*Dec*)

**Input** : Revoked ciphertext $CT'$, user's attribute set $S_{DU}$, CS's key $sk_{CS,DU}$, user's key $sk_{DU}$, the verification tag from DG $tag_{DG}$

**Output:** Message $m$ or $\perp$

1   **CS side:**
2   **if** $S_{DU}$ *does not satisfy* $(M', \tau')$ **then**
3     **return** $\perp$ ;
4   **else**
5     Define $A' = \{i \mid \tau'(i) \in S_{DU}\} \subset \{1, \dots, d'\}$ ;
6     Find constant elements $\eta'_i \in \mathbb{Z}_p^*$ such that $\sum_{i \in A'} \eta'_i \cdot M'_i = (1, 0, \dots, 0)$ ;
7     Calculate $(W^2, C_{DU}, D_{DU})$ by Equation (6);
8     Send the partially decrypted ciphertext $CT_{part} = (C'_1, C_{DU}, D'_1, D_{DU}, tag')$ to the DU ;

9   **DU side:**
10   **if** $tag' \neq tag_{DG}$ **then**
11     **return** $\perp$ ;
12   **else**
13     Compute $m$ and $m''$ by Equation (7);
14   **if** $\overline{tag}' \neq \Phi^{H(m)} \Psi^{H(m'')}$ **then**
15     **return** $\perp$ ;
16   **else**
17     **return** $m$

---

Correctness. Now, we clarify the correctness of our presented solution. In the *Dec* algorithm, it holds that

$$
\begin{aligned}
W^2 &= \prod_{i \in A'} (\hat{e}(K_0, C'_{2,i}) \cdot \hat{e}(K_{\tau'(i)}, C'_{3,i}))^{2\eta'_i}, \\
&= \prod_{i \in A'} (\hat{e}(g^s, g^{a(\lambda_i + \lambda'_i)} f_{\tau'(i)}^{-(r_i + r'_i)}) \cdot \hat{e}(f_{\tau'(i)}^s, g^{r_i + r'_i}))^{2\eta'_i} \\
&= \prod_{i \in A'} \hat{e}(g^s, g^{a(\lambda_i + \lambda'_i)})^{2\eta'_i} \\
&= \hat{e}(g^s, g^a)^{2 \sum_{i \in A'} \eta'_i (\lambda_i + \lambda'_i)} \\
&= \hat{e}(g, g)^{2as(r+r')},
\end{aligned}
$$

$$C_{DU} = C'_0 / \frac{\hat{e}(C'_1, K)}{W^2}$$

$$= \frac{m \cdot v^{r+r'} \cdot \hat{e}(g,g)^{2as(r+r')}}{\hat{e}(g^{r+r'}, g^{\alpha_1} g^{as})}$$

$$= m \cdot \hat{e}(g^{r+r'}, g^{\alpha_2} g^{as}),$$

$$D_{DU} = D'_0 / \frac{\hat{e}(D'_1, K)}{\hat{e}(D'_2, K_0)^2 \cdot W^2}$$

$$= \frac{m \cdot v^{z+z'+r+r'} \cdot \hat{e}(g^{a(z+z')}, g^s)^2 \cdot \hat{e}(g,g)^{2as(r+r')}}{\hat{e}(g^{z+z'+r+r'}, g^{\alpha_1} g^{as})}$$

$$= m \cdot \hat{e}(g^{z+z'+r+r'}, g^{\alpha_2} g^{as}).$$

Thus, the following equality holds:

$$\frac{C_{DU}}{\hat{e}(C'_1, sk_{DU})} = \frac{m \cdot \hat{e}(g^{r+r'}, g^{\alpha_2} g^{as})}{\hat{e}(g^{r+r'}, g^{\alpha_2} g^{as})} = m$$

$$\frac{D_{DU}}{\hat{e}(D'_1, sk_{DU})} = \frac{m'' \cdot \hat{e}(g^{z+z'+r+r'}, g^{\alpha_2} g^{as})}{\hat{e}(g^{z+z'+r+r'}, g^{\alpha_2} g^{as})} = m''.$$

## 5. Security Analysis

In this section, we first demonstrate the semantic security of the proposed approach through a reduction to the q-parallel BDHE assumption. Then, we give a integrity proof of our proposed scheme by reducing it to the DL assumption.

**Theorem 1.** *Under the q-parallel BDHE assumption, the proposed scheme is selectively secure against unauthorized access.*

**Proof.** Suppose a PPT adversary $\mathcal{A}$ breaks the selective security with an advantage $\epsilon$. We construct a simulator $\mathcal{B}$ that solves the q-parallel BDHE problem. The simulator $\mathcal{B}$ is given the q-parallel BDHE instance $\Delta$ as shown in Equation (1), along with an element $T \in \mathbb{G}_T$, which is either $\hat{e}(g,g)^{a^{q+1}r}$ or a random element. The task of $\mathcal{B}$ is to output 0 if $T = \hat{e}(g,g)^{a^{q+1}r}$ and 1 otherwise. The simulator $\mathcal{B}$ interacts with the adversary $\mathcal{A}$ through the following game:

(1) Init. $\mathcal{A}$ submits a challenge access policy $(M^*_{d^* \times e^*}, \tau^*)$, where $d^*, e^* \leq q$.

(2) Setup. The simulator $\mathcal{B}$ first randomly selects $\alpha', \alpha'_1, \alpha'_2, \Phi, \Psi \in \mathbb{G}$ and a hash function $H : \mathbb{G}_T \to \mathbb{Z}^*_p$ with $\alpha' = \alpha'_1 + \alpha'_2$. Then, it implicitly defines $\alpha_1 = \alpha'_1 + a^{q+1}/2$, $\alpha_2 = \alpha'_2 + a^{q+1}/2$ and $\alpha = \alpha' + a^{q+1}$, such that $v = \hat{e}(g,g)^\alpha = \hat{e}(g^a, g^{a^q})\hat{e}(g,g)^{\alpha'}$. For each attribute $x \in U$, $\mathcal{B}$ computes $f_x = g^{t_x} \prod_{i \in X} \prod_{j=1}^{e^*} g^{a^j M^*_{i,j}/b_i}$, where $X = \{i \mid \tau^*(i) = x\}$, and $t_x$ is randomly selected from $\mathbb{Z}^*_p$. Note that $f_x = g^{t_x}$ if $X = \emptyset$. The public key is published as

$$pp = (g, v, g^a, \Phi, \Psi, H, \{f_x\}_{x \in U}).$$

(3) Phase I. For each private key query on a set $S_u$ that does not satisfy $(M^*, \tau^*)$, $\mathcal{B}$ first constructs a vector $\vec{w} = (w_1, \ldots, w_{e^*})$ with $w_1 = -1$ and $\vec{w} \cdot M^*_i = 0$ for all $i$, where $\tau^*(i) \in S$. Then, it randomly selects $\mu \in \mathbb{Z}^*_p$, implicitly defines $h = \mu + \sum_{j=1}^{e^*} w_j a^{q-j+1}$, and computes

$$K = g^{\alpha'_1} g^{a\mu} \prod_{j=2}^{e^*} (g^{a^{q+2-j}})^{w_j},$$

$$K_0 = g^h,$$

$$K_x = K_0^{t_x} \prod_{i \in X} \prod_{j=1}^{e^*} \left( g^{(\frac{a_j}{b_i})^\mu} \prod_{\substack{k=1, \\ k \neq j}}^{e^*} g^{(\frac{a^{q+1+j-k}}{b_i})^{w\mu}} \right)^{M^*_{i,j}},$$

$$sk_u = g^{\alpha'_2} g^{a\mu} \prod_{j=2}^{e^*} (g^{a^{q+2-j}})^{w_j}.$$

Note that if $X = \varnothing$, $K_x = K_0^{t_x}$. Finally, $\mathcal{B}$ sends $(K, K_0, \{K_x\}_{x \in S_u}, sk_u)$ to $\mathcal{A}$.

(4) Challenge. $\mathcal{A}$ submits $m_0, m_1$. $\mathcal{S}$ flips a coin $b \in \{0,1\}$ and chooses random element $y_2, \cdots y_{e^*}$ to generate the vector $\vec{u} = (r, ra + y_2, ra^2 + y_3, \cdots, ra^{e^*-1} + y_{e^*})$. Then, it randomly selects $z, r_1, r_2 \cdots r_{d^*} \in \mathbb{Z}_p^*, m' \in \mathbb{G}_T, Q \in \mathbb{G}$, defines the set $Y_i = \{j \mid \tau^*(j) = \tau^*(i) \wedge j \neq i\}$ for all $i \in [1, e^*]$, and computes

$$C_0 = m_b \cdot T \cdot \hat{e}(g^r, g^{\alpha'}), \quad C_1 = g^r, \quad C_{2,i} = g^{-r_i - rb_i},$$

$$C_{3,i} = f_{\tau^*(i)}^{r'_i} (g^{b_i r})^{-t_{\tau^*(i)}} \prod_{j=2}^{n^*} g^{aM^*_{i,j} y_j} \prod_{k \in Y_i} \prod_{j=1}^{n^*} (g^{\frac{a^j s b_i}{b_k}})^{M^*_{k,j}},$$

$$D_0 = m' \cdot T \cdot \hat{e}(g^r, g^{\alpha'}) \cdot v^z, \quad D_1 = g^r g^z, \quad D_2 = (g^a)^z.$$

$\mathcal{B}$ sends the challenge ciphertext $CT^* = (C_0, C_1, C_{2,i}, C_{3,i}, D_0, D_1, D_2, Q)$ to $\mathcal{A}$.

(5) Phase II. $\mathcal{A}$ continues private key queries for unauthorized sets as in Phase I.

(6) Guess. $\mathcal{A}$ output a guess $b'$ of $b$. If $b' = b$, $\mathcal{B}$ outputs 0 to guess that $T = \hat{e}(g,g)^{a^{q+1}r}$; otherwise, $\mathcal{B}$ outputs 1.

Except for the challenge phase, the simulation is perfect. Next, we rigorously analyze the simulation of verification tag and the revoked ciphertext. In this phase, $\mathcal{B}$ returns a random element $Q \in \mathbb{G}$ instead of $tag = \Phi^{H(m_b)} \Psi^{H(m')}$. However, the distribution of $Q$ is identical to that of $tag$ because the adversary $\mathcal{A}$ has no knowledge of $m'$. From $\mathcal{A}$'s perspective, $\Phi^{H(m_b)} \Psi^{H(m')}$ and $Q$ are indistinguishable.

When revocation occurs, the distribution of the revoked ciphertext remains identical to the original ciphertext distribution from $\mathcal{A}$'s view. In our scheme, the revoked ciphertext replaces $m'$ with a random element $m'' \in \mathbb{G}_T$. Since $\mathcal{A}$ has no information about $m'$, the values $m'$ and $m''$ are perfectly indistinguishable. Additionally, the revoked ciphertext are the elements in $\mathbb{G}_T$ generated by $\vec{u} + \vec{u'}$, $r_j + r'_j$, and $z + z'$, where $\vec{u'}$, $r'_j$, and $z'$ are randomly selected during revocation, while $\vec{u}$, $r_j$, and $z$ are random parameters from the initial ciphertext. So, $\vec{u} + \vec{u'}$, $r_j + r'_j$, and $z + z'$ remain random to $\mathcal{A}$. In summary, the ciphertexts before and after revocation are indistinguishable to adversary $\mathcal{A}$.

We now analyze the probability that the simulator $\mathcal{B}$ solves the q-parallel BDHE assumption. If $T$ is random, the view of $\mathcal{A}$ is independent of $b$, which implies $Pr[b' = b \mid T$ is random$] = 1/2$. If $T = \hat{e}(g,g)^{a^{q+1}r}$, the output of $\mathcal{B}$ depends on $\mathcal{A}$'s guess. In this case, we have $Pr[b' = b \mid T = \hat{e}(g,g)^{a^{q+1}r}] = 1/2 + \epsilon$. So, $\mathcal{B}$'s advantage in solving the q-parallel BDHE problem is

$$\begin{aligned} \epsilon' &= |Pr[b' = b \mid T = \hat{e}(g,g)^{a^{q+1}r}] - Pr[b' = b \mid T \text{ is random}]| \\ &= |1/2 + \epsilon - 1/2| \\ &= \epsilon. \end{aligned}$$

This indicates that $\mathcal{B}$ can break the q-parallel BDHE assumption with a non-negligible advantage, which contradicts the assumption itself. Consequently, our scheme is proven to be selectively secure against unauthorized accesses under the q-parallel BDHE assumption. □

**Theorem 2.** *Our scheme achieves the data integrity under the DL assumption.*

**Proof.** If an adversary $\mathcal{A}$ can successfully compromise the integrity of the proposed scheme, we can construct a simulator $\mathcal{B}$ capable of solving the DL problem. $\mathcal{B}$ is given a discrete logarithm instance $(\hat{e}, \mathbb{G}, \mathbb{G}_T, p, g, g^\delta)$; its goal is to obtain $\delta$.

(1) Setup. Simulator $\mathcal{B}$ sets a bilinear pairing $(\hat{e}, \mathbb{G}, \mathbb{G}_T, g, p)$ and selects $a, \alpha, \alpha_1, \alpha_2,$ $\eta \in \mathbb{Z}_p, f_1, f_2, \cdots, f_{|U|} \in \mathbb{G}$ as well as a hash function $H : \mathbb{G}_T \to \mathbb{Z}_p^*$. Then, $\mathcal{B}$ sets $v = \hat{e}(g,g)^\alpha, \Phi = g^\delta, \Psi = g^\eta$ and outputs the public parameters $pp = (\hat{e}, \mathbb{G}, \mathbb{G}_T, g, p, \Phi, \Psi,$ $v, g^a, H, \{f_x\}_{x \in U})$ to $\mathcal{A}$.

(2) Phase I. When $\mathcal{A}$ issues a private key query for an attribute set $S$, since $\mathcal{B}$ possesses the master secret key, $\mathcal{B}$ is able to generate the corresponding private keys and return them to $\mathcal{A}$.

(3) Challenge. $\mathcal{A}$ chooses a message $m$ as well as the access policy $(M, \tau)$, and sends them to the challenger $\mathcal{B}$. $\mathcal{B}$ executes the $Enc_{DO}$ algorithm to obtain the ciphertext $CT_{init} = ((M, \tau), C_0, C_1, D_0, D_1, D_2, tag, \{\lambda_i\}_{i=1}^d)$, where $tag = \Phi^{H(m)}\Psi^{H(m')}$.

**Case 1** (no revocation). *$\mathcal{B}$ returns $CT^* = CT_{init}$ to $\mathcal{A}$.*

**Case 2** (with revocation). *$\mathcal{B}$ executes Delegate to obtain $DG = ((\widetilde{M}, \widetilde{\tau}), \overline{D}_0, tag')$, where $tag' = \Phi^{H(m)}\Psi^{H(m'')}$. Then, $\mathcal{B}$ sends $CT^* = (CT_{init}, DG)$ to $\mathcal{A}$.*

(4) Phase II. $\mathcal{A}$ continues to issue queries as in Phase I, and $\mathcal{B}$ responds the queries as in Phase I.

(5) Output. $\mathcal{A}$ outputs a partially decrypted ciphertext $CT_{part}$. When there is no revocation, the ciphertext contains the verification tag $\overline{tag}$; when revocation occurs, the ciphertext contains the verification tag $\overline{tag}'$ (we set the updated access policy as $(M', \tau')$).

**Case 1** (no revocation). *Simulator $\mathcal{B}$ selects an attribute set $S_u$ that satisfies $M$, generates the corresponding private key $sk_u$, and decrypts the ciphertext $CT_{part}$ using this key to obtain the plaintext $\overline{m}$ and random message $\overline{m}'$. If $\mathcal{A}$ wins the integrity game, then $\overline{m} \notin \{m, \perp\}$ (i.e., $\overline{m} \neq m$ and $\overline{tag} = tag$). Therefore, $\mathcal{B}$ has*

$$\overline{tag} = tag$$
$$\Leftrightarrow \Phi^{H(\overline{m})}\Psi^{H(\overline{m}')} = \Phi^{H(m)}\Psi^{H(m')}$$
$$\Leftrightarrow g^{\delta \cdot H(\overline{m}) + \eta \cdot H(\overline{m}')} = g^{\delta \cdot H(m) + \eta \cdot H(m')}$$
$$\Leftrightarrow \delta \cdot H(\overline{m}) + \eta \cdot H(\overline{m}') = \delta \cdot H(m) + \eta \cdot H(m').$$

*Finally, $\mathcal{B}$ derives $\delta = \frac{\eta \cdot (H(m') - H(\overline{m}'))}{H(\overline{m}) - H(m)}$ and outputs $\delta$ as its answer for the DL assumption.*

**Case 2** (with revocation). *$\mathcal{B}$ selects the attribute set $S_u'$ that satisfies the updated policy $M'$, generates the corresponding private key, and decrypts $CT_{part}$ to obtain the plaintext $\overline{m}$ and the random message $\overline{m}''$. If $\mathcal{A}$ wins the game, this implies $\overline{m} \notin \{m, \perp\}$. Therefore, $\mathcal{B}$ obtains $\overline{tag}' = tag'$. Then, $\mathcal{B}$ computes $\delta = \frac{\eta \cdot (H(m'') - H(\overline{m}''))}{H(\overline{m}) - H(m)}$ as the answer to the DL assumption.*

Analysis. The simulation is perfectly executed. As demonstrated in Case 2, our scheme still achieves the data integrity even after the tag has been updated. The advantage of $\mathcal{B}$ in breaking the DL problem is precisely equivalent to the advantage of $\mathcal{A}$ in winning the integrity game. $\square$

# 6. Performance Analysis

This section systematically compares our proposed scheme with the methods proposed by Ge et al. [7] and Huang et al. [39] using theoretical analysis and experimental tests. The comparison focuses on computational overhead and functional characteristics to validate the comprehensive performance advantages of our scheme in smart health environments.

## 6.1. Theoretical Analysis

First, for clarity, we define $T_p$, $T_e$, $T_s$ as the time needed for one bilinear pairing operation, one exponentiation operation in $\mathbb{G}_T$, and one scalar multiplication in $\mathbb{G}$, respectively. Let $b$ and $c$ represent the row quantity in the access control matrix and the number of attribute, respectively. Table 1 presents a comparison of computational costs across different phases. In terms of key generation, our scheme incurs slightly higher costs than [7] but outperforms [39] in efficiency. Unlike prior schemes that perform all encryption tasks on the DO side, our approach splits encryption between the DO and CS. The total encryption cost of our scheme is lower than [7] but higher than [39] while the DO-side cost remains constant at $2T_e + 5T_s$, which is significantly lower than the non-constant costs of both [7] and [39]. For revocation, our costs are lower than [7] but higher than [39]. Like the encryption algorithm, in contrast to existing schemes that execute all decryption tasks on the DU side, we distribute decryption between the DU and CS. The total decryption cost of our scheme is lower than [7] but higher than [39] while the DU-side cost remains constant at $2T_p + 2T_s$, which is notably lower than the variable costs of the other two schemes. Overall, our scheme achieves constant and low local computation costs, and the moderately higher CS-side costs for revocation and partial encryption/decryption operations are justified by the CS's superior computational capacity. This design leverages the powerful processing capabilities of the cloud to offload complex computations and ensures that the local costs remain constant and low, which is critical for resource-constrained IoT medical devices in smart health.

**Table 1.** Comparison of computational cost.

| Scheme | Key Generation | Encryption | Revocation | Decryption |
|--------|----------------|------------|------------|------------|
| [7] | $(c+3)T_s$ | $2T_e + (4+6b)T_s$ | $2T_e + (2+6b)T_s$ | $(2+4c)T_p + 2cT_e + 2T_s$ |
| [39] | $(c+8)T_s$ | $T_e + (3+3b)T_s$ | $T_e + (1+3b)T_s$ | $(1+2c)T_p + cT_e + 2T_s$ |
| Ours | $(c+5)T_s$ | DO: $2T_e + 5T_s$<br>CS: $3bT_s$ | $2T_e + (3+3b)T_s$ | CS: $(2c+3)T_p + (c+1)T_e$<br>DU: $2T_p + 2T_s$ |

Table 2 highlights functional advantages. Our scheme ensures dynamic updates of the verification tag, which prevents revoked users from inferring plaintext via tag analysis. In contrast, the verification tags in the other two schemes remain unchanged during revocation, which poses security risks. Furthermore, our scheme employs LISS for access structure. Unlike LSSS over finite fields, LISS leverages secret sharing over integers and reduces the cost of secret sharing. These advantages demonstrate the practicality of our scheme for large-scale collaborative IoT medical networks.

**Table 2.** Comparison of functionality.

| Scheme | Revocable | Verifiable | Tag Update | Access Policy |
|:------:|:---------:|:----------:|:----------:|:-------------:|
| [7] | ✓ | ✓ | ✗ | LSSS |
| [39] | ✓ | ✓ | ✗ | LSSS |
| Ours | ✓ | ✓ | ✓ | LISS |

*6.2. Experimental Tests*

To evaluate computational efficiency, we designed an experimental system, which we describe in the following four aspects: the hardware platform, cryptographic implementation, security parameters, and experimental protocol. The hardware platform is based on a 2.4-GHz Intel i5 processor that runs the Windows XP operating system. Cryptographic operations leverage the Stanford Pairing-Based Cryptography (PBC) library [41], which enables the implementation of the Tate pairing $\hat{e} : \mathbb{G} \times \mathbb{G} \to \mathbb{G}_T$ over the supersingular elliptic curve $E/\mathbb{F}_p : y^2 = x^3 + x$. In this context, $\mathbb{G}$ denotes a group of points on $E/\mathbb{F}_p$ and $\mathbb{G}_T$ represents a subgroup of $\mathbb{F}_{p^2}$. To achieve security equivalent to 1024-bit RSA, critical parameters are configured as follows: the embedding degree of the curve is set to 2, and the prime $p$ is selected such that $p \equiv 3 \mod 4$. Finally, the runtime of three core cryptographic operations (bilinear pairing, exponentiation and scalar multiplication) is measured by repeating the experiments 1000 times on an Intel-core personal computer. This protocol guarantees statistical reliability by minimizing random fluctuations. According to [42], in this experiment, the value of $T_p$, $T_e$, and $T_s$ are 25.1 ms, 2.84 ms, and 11.88 ms, respectively. To simulate realistic smart healthcare deployment scenarios, we established the experimental parameters as follows: The number of user attributes ranged from 5 to 50 with an increment of 10. During original ciphertext generation, the access policy $\mathcal{T}$ took the form of a logical AND gate over selected attributes. For the revocation algorithm, the revocation access policy $\widetilde{\mathcal{T}}$ varied from 5 to 50 attributes with an increment of 5. This configuration produced a revoked access policy $\mathcal{T}' = \mathcal{T} \ AND \ \widetilde{\mathcal{T}}$ whose dimension spanned from 10 to 100 with an increment of 10.

Through comparative analysis of experimental results, we can clearly observe the significant computational efficiency advantages of our proposed scheme. Figure 3 demonstrates the trend of local encryption computational overhead with varying access policy dimensions, where our scheme maintains constant computational costs as dimensions increase from 5 to 50, while the schemes by Ge et al. and Huang et al. exhibit clear linear growth patterns. In the comparison of revocation computational overhead shown in Figure 4, although all three schemes demonstrate linearly increasing computational costs with growing dimensions, our scheme and Huang et al.'s approach show lower growth slopes compared to Ge et al.'s method. Notably, as shown in Figure 4, the performance curves of our scheme and Huang et al.'s scheme nearly overlap, yet the computational cost of our scheme is marginally higher. This slight efficiency compromise is entirely justified, as our scheme achieves enhanced security through a verifiable tag update mechanism, which demonstrates a deliberate and reasonable trade-off between security and efficiency. Figure 5 specifically compares the trends of local decryption overhead with increasing attribute quantities. Experimental data confirm that our scheme maintains consistently low and stable decryption overhead regardless of attribute quantity growth.

The experimental results from all three datasets exhibit strong consistency with theoretical analysis. The experimental outcomes conclusively prove that our solution achieves an optimal balance between security assurance and computational efficiency for smart health systems.

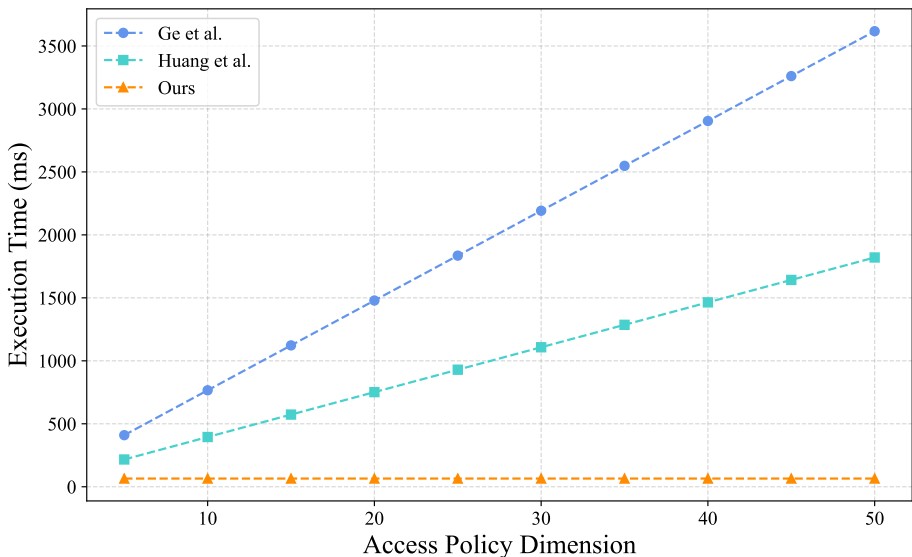

**Figure 3.** Comparison of local encryption cost [7,39].

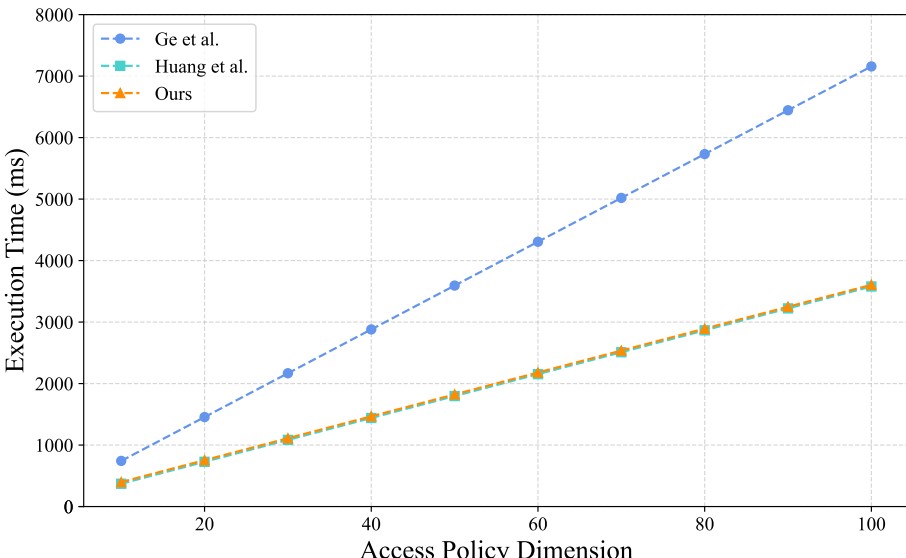

**Figure 4.** Comparison of revocation cost [7,39].

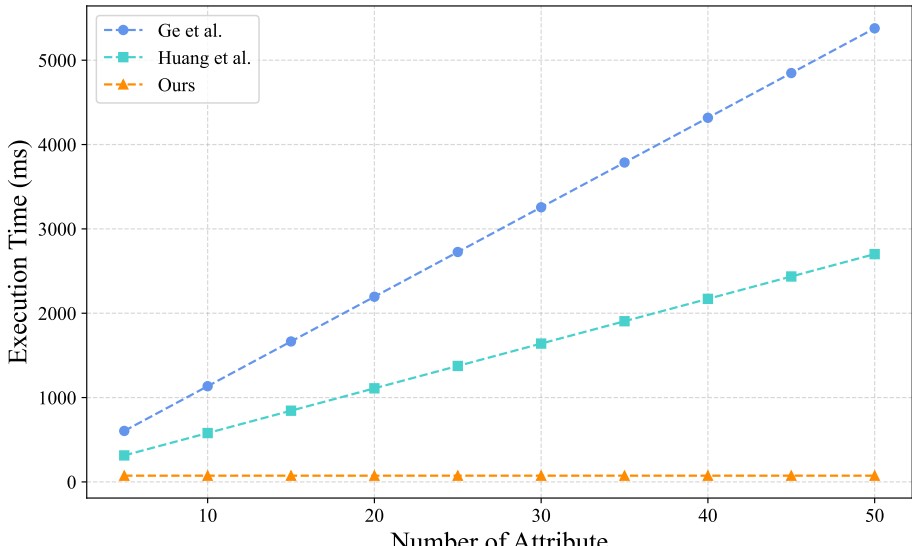

**Figure 5.** Comparison of local decryption cost [7,39].

## 7. Future Work

To further improve the practical applicability and security robustness of our proposed scheme, future work will explore the following directions:

- Decentralized KGC Architectures: The current scheme relies on a single KGC, which introduces risks such as single points of failure and centralized trust. To address these limitations, the framework can be extended to multi-authority KGC architectures or threshold-based KGC architectures. The former distributes trust among independent authorities, which eliminates centralized control and improves compatibility with decentralized healthcare ecosystems. The latter employs $(t, n)$-threshold cryptography to ensure that key generation requires collaboration from at least $t$ out of $n$ authorities. This design prevents compromises even if some authorities are corrupted.

- Security Model Strengthening: While the current scheme achieves selective security under static adversarial assumptions, it cannot fully resist attacks from adversaries who adaptively select attack targets during security games. To overcome this limitation, the scheme must be upgraded to achieve full adaptive security, which would significantly enhance its resilience in dynamic real-world environments.

To implement these extensions, the following critical challenges must be carefully addressed and resolved:

- For decentralized KGC architectures, secure and efficient distributed key generation protocols must be designed to balance cryptographic robustness with computational overhead. This requires establishing reliable communication channels among multiple KGC nodes to prevent eavesdropping and tampering. Furthermore, dynamic scenarios such as nodes join/exit demand adaptable protocols to maintain stability without sacrificing security.

- For achieving full adaptive security, we must redesign the scheme to include a hash function that maps bit strings to points in group $\mathbb{G}$. Due to the computational overhead of this function, it is critical to rigorously define where and how it should be applied. Furthermore, the security proof must be reformulated under a stronger adversarial model where attackers adaptively query secrets and issue challenges. Finally, even with the increased complexity of addressing adaptive attacks, the scheme must maintain computational efficiency to ensure practical deployment in real-world scenarios.

Through these improvements, the proposed scheme can achieve full adaptive security, adapt to more sophisticated decentralized application scenarios, and preserve the functional advantages of the existing design. Theoretical analysis and practical implementation of these extensions will form the core focus of our future research.

## 8. Conclusions

This study proposes a revocable and verifiable ABE scheme tailored for time-sensitive and secure transmission requirements in smart health systems. The scheme uses an attribute-based encryption framework to achieve dynamic fine-grained access control and provides efficient verification mechanism. Crucially, it enforces strict ciphertext update to ensure that revoked users cannot extract any useful information from the previous ciphertexts. This resolves the risk in existing schemes where revoked users could infer plaintext through unchanged verification tags. We formally prove the semantic security and integrity of the scheme and conduct simulations to evaluate its performance. The results demonstrate that our scheme effectively addresses data transmission challenges in smart health systems and is highly suitable for such environments.

**Author Contributions:** Conceptualization, Z.C. and L.H.; Investigation, Z.C. and L.H.; Writing—original draft preparation, Z.C.; Writing—review and editing, L.H. and B.H. All authors have read and agreed to the published version of the manuscript.

**Funding:** This work was supported in part by the Hangzhou Joint Fund of the Zhejiang Provincial Natural Science Foundation of China under Grant LHZSZ24F020002, and in part by the National Natural Science Foundation of China under Grant U21A20466.

**Data Availability Statement:** Data are contained within the article.

**Conflicts of Interest:** The authors declare no conflicts of interest.

# Appendix A

*Appendix A.1. Linear Secret Sharing Scheme (LSSS)*

Let $U$ denote the universal set of attributes that contains $n$ types of attributes. Matrix $A$ is an $l \times n$ matrix over $Z_p$ that serves as the generation matrix $A(l \times n)$ in the LSSS. The injective function $\rho$ maps each row of matrix $A$ to an attribute index. This matrix $A$ defines the structure of the access control policy.

A linear secret sharing scheme can be constructed when the following two conditions are satisfied:

- Secret Distribution: Consider the vector $v = (s, y_2, \ldots, y_n)$, where $s \in \mathbb{Z}_p$ is the shared secret and $y_2, \ldots, y_n \in \mathbb{Z}_p$ are random numbers used to conceal $s$. Each secret share is computed as $\lambda_i = A_i \cdot v$, where $A_i$ denotes the $i$-th row of matrix $A$. The share $\lambda_i$ corresponds to the $i$-th portion of the secret $s$, which is mapped to an authorized attribute index by the injective function $\rho$.
- Secret Reconstruction: For any authorized attribute set $S \in A$ in the access structure, define $I = \{i \mid \rho(i) \in S\}$. There exists a polynomial-time algorithm that generates coefficients $\{\omega_i \in \mathbb{Z}_p\}_{i \in I}$ based on the generation matrix $A$, which satisfy $\sum_{i \in I} \omega_i A_i = (1, 0, \ldots, 0)$. Consequently, the secret value $s$ can be reconstructed through $s = \sum_{i \in I} \omega_i (A_i \cdot v) = \sum_{i \in I} \omega_i \lambda_i$. For unauthorized sets, no valid coefficients exist, and thus, the secret cannot be recovered.

*Appendix A.2. Performance Comparison Between LISS and LSSS*

(1) Theoretical Analysis. The LSSS scheme is based on the finite field $\mathbb{Z}_p$, which requires modular arithmetic operations. When $p$ is large, these modular operations can significantly increase the computational overhead. In contrast, LISS operates over the integer domain without requiring any modular arithmetic. Furthermore, the parameters of LISS (such as the range of random numbers) can be flexibly adjusted according to security requirements, which avoids excessive redundancy.

(2) Experimental Evaluation. To further evaluate the efficiency of LISS and LSSS in secret distribution and reconstruction, we conducted an experiment on a Windows 11 system with an Intel(R) Core(TM) i5-8250U CPU @ 1.60GHz. The parameters were set as follows: $k = 32$, prime $p = 2^{61} - 1$, and $l = 5$. For secret distribution, the number of shareholders ranged from 5 to 20 in increments of 5. For reconstruction, authorized attribute sets of sizes 2 to 8 (in steps of 2) were tested. The results are shown in Figures A1 and A2. Figure A1 compares the execution time of LISS and LSSS during secret distribution, while Figure A2 highlights their performance in reconstruction. Both figures clearly demonstrate that LISS outperforms LSSS in terms of efficiency, which aligns with our theoretical analysis.

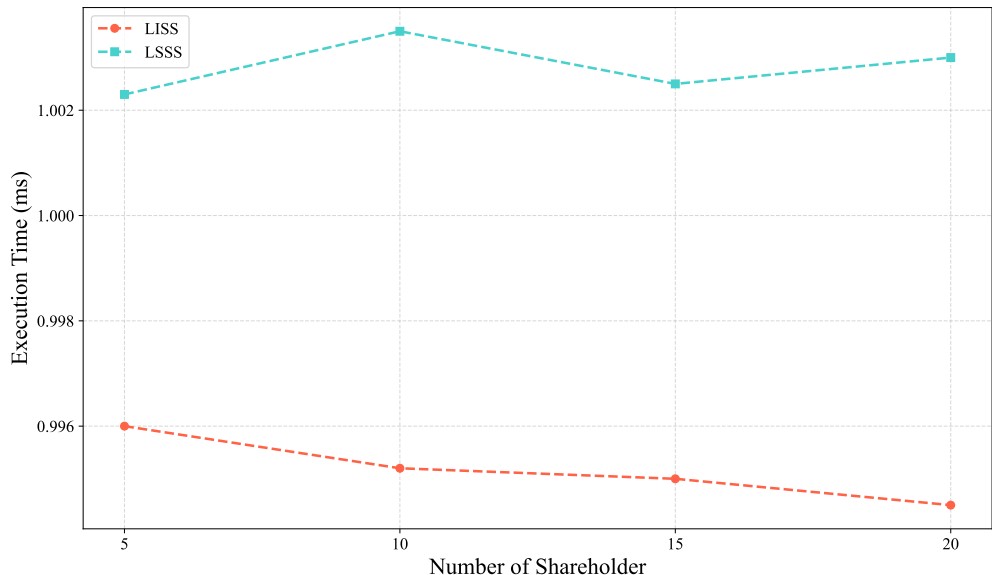

**Figure A1.** Comparison of secret distribution.

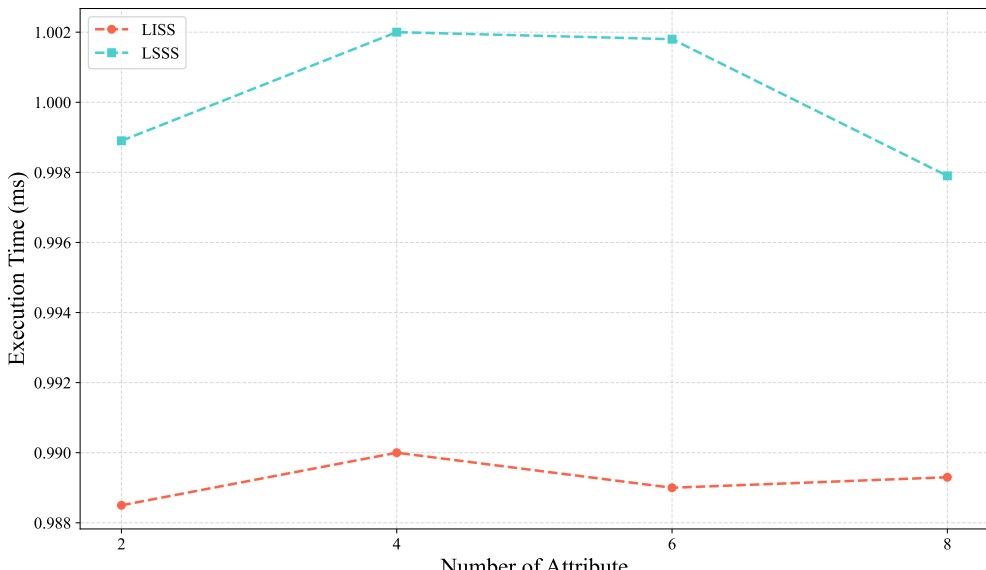

**Figure A2.** Comparison of secret reconstruction.

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
