# Peer review of "Revocable Attribute-Based Encryption with Efficient and Secure Verification in Smart Health Systems"

_mathematics, doi:10.3390/math13091541_

Round 1
Reviewer 1 Report
Comments and Suggestions for Authors
The authors have presented a revocable attribute based encryption scheme with secure verification for smart healthcare systems.
The proposed work may be appreciable. However, there are few concerns needed to be addressed before publication.
1. The authors need to emphasis on the core cryptographic algorithm performed by DO. The authors may even include a pseudocode for better readability and
understanding.
2. The strength of algorithm is dependent on the strength of the key. The KDC has generated two key sets msk and Su by picking a random element s ∈ Z.
2.1. The authors should validate the selection of s from Z space.
2.2. The authors should check the sensitivity of key and evaluate the key space. The can also perform Brute force attack analysis.
2.2. The manuscript does not contain information on key distribution algorithm from KDC to DU and CS which is more crucial.
3. In performance comparison, the authors have calculated both theoretical and experimental computational time cost. The authors should also include
dataset size or system configurations for validating the performance claims.
4. The authors should also include comparison of proposed ABE scheme in terms of efficiency and revocation latency
Reviewer 2 Report
Comments and Suggestions for Authors
This paper presents a novel Revocable Attribute-Based Encryption (RABE) scheme tailored for smart healthcare systems, where both real-time efficiency and strong security guarantees are required. The scheme tackles two important limitations in prior works: (1) inefficient verification due to duplicate encryption of auxiliary messages, and (2) security vulnerability stemming from static verification tags post-revocation. The authors address these challenges through a refined encryption architecture, the adoption of Linear Integer Secret Sharing (LISS), and a carefully designed tag update mechanism during revocation.
The manuscript is generally well-structured, the technical depth is sufficient, and the experimental validation is compelling. The theoretical security proofs are built upon standard assumptions and are rigorously presented. Nonetheless, there are some areas that merit clarification or improvement.
Weaknesses and Concerns:
- Assumptions on Trust and Threat Model:
The system assumes the Key Generation Center (KGC) is fully trusted, which might be too strong for some practical deployments (e.g., decentralized healthcare systems). It would enhance the completeness of the work to briefly discuss potential extension to multi-authority or threshold KGC models.
- Lack of Formal Privacy Analysis for the Outsourced Server:
Although the cloud server is considered "semi-honest", the scheme does not explicitly quantify the privacy leakage (if any) from partial decryption. For example, could tag patterns or partial ciphertexts be used to infer access patterns or data usage?
- Verification Tag Structure:
The verification tag is constructed as tag=??, which is elegant. However, the scheme assumes the hash function is collision-resistant and injective over GT, which may not be realistic in practice. The paper should discuss how potential hash collisions are avoided or mitigated.
- Choice of LISS over LSSS:
The paper rightly claims that LISS reduces overhead compared to LSSS, but experimental evidence on actual performance differences in secret sharing is lacking. A small benchmark comparing LISS vs. LSSS secret distribution and reconstruction would strengthen the argument.
- Security Model Limitations:
The scheme assumes a selective security model. While this is common in ABE works, it is generally weaker than adaptive security. The authors may consider discussing the gap and whether their scheme could be extended to support full adaptive security.
Minor Comments and Suggestions:
- Figure 2 lacks sufficient explanation
The system architecture diagram is relatively brief. It is recommended to expand the figure caption with a concise description of the interaction flows, particularly between the Cloud Server (CS), Data Owner (DO), and Data User (DU).
2. Experimental parameter setting could be more realistic
The system evaluation could be enriched by simulating a more realistic number of attributes or users, especially considering practical hospital deployments. The current evaluation varies the dimension of the access policy and number of attributes from 10 to 100. It would be more convincing to simulate scenarios closer to real-world smart healthcare deployments—for example, with 100 physician users and 20 distinct attributes.

Round 2
Reviewer 2 Report
Comments and Suggestions for Authors
After carefully reviewing the authors’ revised manuscript and response to the first-round comments, I find that the authors have not adequately addressed several key technical concerns, and their response lacks the expected rigor and transparency. My detailed comments are as follows:
1.Lack of Empirical Comparison between LISS and LSSS
The previous review explicitly requested experimental comparison between LISS and LSSS in terms of computational efficiency during secret distribution and reconstruction. However, the authors only referenced the theoretical analysis from Reference [21] without presenting any empirical data, numerical benchmarks, or even minimal comparative plots. Deferring this to "future work" is not acceptable in a technical journal that demands evidence-based validation.
Recommendation: A basic empirical comparison between LISS and LSSS should be added, even in the appendix or as simplified results.
2.No Substantive Response Regarding Selective vs. Adaptive Security
In response to the comment on the security model, the authors simply restate the conceptual difference between selective and adaptive security and justify their use of the former based on efficiency. However, they do not discuss any potential methods for extending the scheme to adaptive security, nor do they propose lightweight cryptographic mechanisms or future-proof design pathways. Moreover, no new material has been added to the manuscript regarding this important limitation.
Recommendation: At minimum, the authors should include a paragraph in the manuscript discussing the technical challenges and potential design strategies for supporting adaptive security.
3. Non-standard Format of the Response Letter and Lack of Revision Transparency
The response letter does not provide side-by-side comparisons or summaries of the revised content. References such as “see Page X, Line Y” are too vague and require reviewers to manually locate and verify every change. This approach violates the best practices of transparent scholarly communication. Revision responses must clearly indicate the original text, the modified version, and where the change occurs.
Recommendation: A revised response letter should include structured summaries of each modification, ideally with quotations or comparisons of before-and-after text.
Round 3
Reviewer 2 Report
Comments and Suggestions for Authors
Accept in present form.